# The Use of Pulse Oximetry in the Assessment of Acclimatization to High Altitude

**DOI:** 10.3390/s21041263

**Published:** 2021-02-10

**Authors:** Tobias Dünnwald, Roland Kienast, David Niederseer, Martin Burtscher

**Affiliations:** 1Institute for Sports Medicine, Alpine Medicine and Health Tourism (ISAG), UMIT—Private University for Health Sciences, Medical Informatics and Technology, 6060 Hall in Tirol, Austria; tobias.duennwald@umit.at; 2Department of Biomedical and Health Technology, Federal Higher Technical Institute for Education and Experimentation—HTL Anichstraße, 6020 Innsbruck, Austria; r.kienast@tsn.at; 3Department of Cardiology, University Hospital Zurich, University Heart Center Zurich, University of Zurich, 8091 Zurich, Switzerland; david.niederseer@usz.ch; 4Department of Sport Science, University of Innsbruck, 6020 Innsbruck, Austria

**Keywords:** pulse oximetry, high altitude, acclimatization, rest and exercise, acute mountain sickness

## Abstract

*Background*: Finger pulse oximeters are widely used to monitor physiological responses to high-altitude exposure, the progress of acclimatization, and/or the potential development of high-altitude related diseases. Although there is increasing evidence for its invaluable support at high altitude, some controversy remains, largely due to differences in individual preconditions, evaluation purposes, measurement methods, the use of different devices, and the lacking ability to interpret data correctly. Therefore, this review is aimed at providing information on the functioning of pulse oximeters, appropriate measurement methods and published time courses of pulse oximetry data (peripheral oxygen saturation, (SpO_2_) and heart rate (HR), recorded at rest and submaximal exercise during exposure to various altitudes. *Results*: The presented findings from the literature review confirm rather large variations of pulse oximetry measures (SpO_2_ and HR) during acute exposure and acclimatization to high altitude, related to the varying conditions between studies mentioned above. It turned out that particularly SpO_2_ levels decrease with acute altitude/hypoxia exposure and partly recover during acclimatization, with an opposite trend of HR. Moreover, the development of acute mountain sickness (AMS) was consistently associated with lower SpO_2_ values compared to individuals free from AMS. *Conclusions*: The use of finger pulse oximetry at high altitude is considered as a valuable tool in the evaluation of individual acclimatization to high altitude but also to monitor AMS progression and treatment efficacy.

## 1. Introduction

Wearable sensors can provide athletes, coaches, patients and physicians with useful physiological data, e.g., on the actual cardiovascular and respiratory stress of the individual [1,2,3]. Such information may be gathered by continuous or spot measurements depending on specific objectives. For example, the monitoring of bio-vital markers like heart rate and peripheral oxygen saturation has become a standard of patient care [4] but is also frequently applied by people visiting high altitudes for sight-seeing, trekking, skiing or climbing [5,6]. Simple and inexpensive devices, finger pulse oximeters, are widely used to monitor physiological responses to high-altitude exposure, the progress of acclimatization, and/or the potential development of high-altitude related diseases [7,8,9]. Although there is increasing evidence for the usefulness of pulse oximetry at high altitude some controversy remains [10,11]. This is largely due to differences in individual preconditions, different evaluation purposes, different measurement methods, not considering limitations of devices in certain conditions, the use of different devices, and the lacking ability to interpret data correctly [12]. Therefore, this review is aimed at providing information on the functioning of pulse oximeters, appropriate measurement methods and published time courses of pulse oximetry data (peripheral oxygen saturation, SpO_2_; and heart rate, HR) recorded at rest and submaximal exercise during exposure to various altitudes. Moreover, where available, alterations in acute mountain sickness (AMS) scores for the particular exposure will be illustrated. Interpretation notes are intended to enable the reader to differentiate between normal and potentially pathologic values recorded by pulse oximetry in consideration of different individual conditions.

## 2. Methods

Part 1: In the first part of this review, we present basic principles of functioning and, based on a selected literature review, most relevant pitfalls and possible countermeasures for pulse oximetry particularly concerning healthy people going to high altitudes.

Part 2: In order to show the various changes in SpO_2_ and HR observed following acute exposures to altitude, the second part of this review presents the results of a literature search performed in the PubMed database including studies published prior to June 2020, using the search terms “oxygen saturation”, “heart rate”, “high altitude”, “acclimatization”, “exercise”, “performance” and “acute mountain sickness”. No restriction was done regarding to the type of studies. The search was complemented by articles known to the authors and by screening reference lists of selected review articles. Studies reporting SpO_2_ data from subjects acutely exposed to high altitude with subsequent active or passive stays were included in this review. Exclusion criteria included the absence of reporting baseline data on SpO_2_, the presentation of data obtained at sea level and once at altitude only, stays at higher altitude levels during the exposure, administration of medications as well as stays lasting less than 2 days and more than 28 days. Core findings of the presented literature overview are summarized. Figure 1 shows a flow chart of the selection process. Studies predominantly involved an observational study design.

Part 3: In the third part, physiologic and pathophysiologic mechanisms explaining pulse oximetric measures when acutely exposed to high altitude and during acclimatization are discussed, including practical aspects for the mountaineer going to high altitude.

## 3. Part 1: Basic Principles of Functioning, Most Relevant Pitfalls and Possible Countermeasures for Pulse Oximetry Particularly Concerning Healthy People Going to High Altitudes

The introduction of pulse oximetry represents one of the most important technological advances in medicine, permitting to continuously, non-invasively and simultaneously monitor SpO_2_ of hemoglobin in the arterial blood and HR. These vital parameters provide exceptionally important information on well-being of the individual, e.g., patients in the hospital setting, in emergency situations at home or in the field, but also in people going to high altitudes.

Hemoglobin (Hb) is a prominent protein complex in erythrocytes. Based on its ability to bind oxygen (O_2_) it is essential for the transport of O_2_ from the alveoli to tissues. Hb exists in two forms: (1) the deoxyhemoglobin (HHb) without attached O_2_ and (2) the oxyhemoglobin (O_2_Hb) with bound O_2_ molecules. O_2_ molecules change the light absorption of Hb at specific wave lengths [6,13]. 

This effect can be observed even under normal light conditions since well oxygenated (arterial) blood with high O_2_Hb concentration exhibits a bright red staining, while venous (deoxygenated) blood appears dark red for the eye. Pulse oximetry makes advantage of this optical effect of diverging light absorption of Hb and utilizing red light at wavelength of 660nm and near-infrared (IR) light at 940 nm to estimate SpO_2_. The distinct feature of these two wavelengths is that red light is more strongly absorbed by HHb than by O_2_Hb, whereas infrared light exhibits the opposite characteristic (see Figure 2). For both wavelengths, the absorption during systole ($A_{AC}$) and diastole ($A_{DC}$) is measured and the modulation ratio R is calculated, where R is expressed as: (1)$$R=\frac{A_{AC,red}/A_{DC,red}}{A_{AC,IR}/A_{DC,IR}}$$

Based on a comparison of R to an empirically generated calibration curve, the SpO_2_ value is estimated [13,14,15]. The data for this calibration curve was acquired from adult healthy volunteers and includes saturation values from 70% to 100% [12,13,15,16,17]. By utilizing multiple wavelengths, selected instruments are also capable of determining the most important dyshemoglobins—carboxyhemoglobin (SpCO) and methemoglobin (SpMet)—in addition to oxy- and deoxyhemoglobin [14,18,19].

Sensors commonly used in pulse oximetry can be categorized into two types, according to the measurement method: (1) Transmission sensors, where the emitter and receiver are placed opposite to each other and the light passes through the tissue (e.g., finger- or earlobe sensors), and (2) reflection sensors, where the emitter and receiver are placed next to each other and the backscattered light is analyzed (e.g., forehead- or wrist sensors) [14]. Because of the easy-to-use and wide clinical acceptance, pulse oximetry has become a well-established indirect method for continuous and noninvasive monitoring of blood oxygenation [21]. By default, medical pulse oximeters comply with the international standard for pulse oximeter manufacture ISO 80601-2-61 and must maintain high accuracy (average root mean square error A_rms_ ≤ 4%) in SpO_2_ reading within the range of 70–100% when compared to arterial oxygen saturation (SaO_2_) obtained from arterial blood gas (ABG) analysis. The FDA recommends A_rms_ values of ≤3.0% for transmission sensors and ≤3.5% for ear clip and reflectance sensors [22]. This also implies that not each oximeter would provide exactly the same reading if theoretically measured in the same individual, at the same time and at the same location. However, the variance should be within the limits specified. Assuming faultless handling of the instrument, these variations are attributable to technical variations, e.g., different signal averaging times, incorrect calibration or differences in the number and precision of wavelengths used [23]. Devices for non-medical use not conforming to this ISO standard may have larger deviations in SpO_2_ readings. There exist a limited number of studies comparing reasonably priced commercially available handheld devices to medical-standard-devices. These studies indicate that “low-cost” handheld devices provide sufficiently accurate SpO_2_ values in the range of about 90–100% compared to medical devices, however, below 90%, non-medical devices decrease in accuracy [16,24,25,26]. This property might be a drawback for measurements at high altitudes. Although, no explicit statement for the use of non-medical devices at high altitudes can be made based on these few studies. In general, there is a lack of data on the measurement accuracy of pulse oximeters at high altitudes comparing SpO_2_ to SaO_2_ obtained from ABG analysis. This complicates to identify devices that are appropriate for their usage at high altitudes without restrictions. However, certain parameters may require increased attention when the decision for a specific pulse oximeter has to be made: (1) accuracy, precision, and bias of the device [12]; (2) environmental conditions such as maximum operating altitude respectively minimum air pressure or the minimum operating temperature; (3) the availability of advanced algorithms to reduce motion artifacts or the detection of low perfusion; (4) the selection of the sensor location, typically using the finger [12,15,27,28]—however, other common positions such as the forehead may be considered, in particular if the measurement is conducted during motion [28,29,30]; (5) if required: the opportunity to sense carboxyhemoglobin and methemoglobin. In addition to the technical aspects, however, the measurement process itself becomes more error-prone with increasing altitude [6,12]. Table 1 summarizes the most significant pitfalls that may lead to inaccurate SpO_2_ readings and possible countermeasures particularly for healthy people visiting high altitudes. The application of standardized procedure by trained users can prevent incorrect measurements and data interpretation [6,12]. Unfortunately, there exists no uniform standard for measuring protocols at high altitudes but there are some specific recommendations available to minimize measurement uncertainties which largely coincides with the countermeasures listed in Table 1: (1) The test person should remain in a sitting position for about 5 min, (2) the measuring site (normally the finger) should be kept as warm as possible, e.g., by wearing gloves, (3) motions of the sensor should be prevented, (4) the sensor should be shielded from ambient light, (5) a trained and experienced examiner should perform the measurements, and (6) SpO_2_ values should be monitored and averaged over a period of 2–3 min [6,12,31]. Additionally, if the device offers the ability to display the pulse wave graphically, it should be ensured that it remains as stable as possible [31]. In the following section, studies implementing a pulse oximeter to document the progress of acclimatization are illustrated. 

## 4. Part 2: Results from the Literature Review 

### 4.1. Resting SpO_2_ and HR Changes during Acclimatization to High Altitude

The findings from 18 studies reporting resting SpO_2_ data are presented in Table 2. Two studies were performed at an altitude of 2000–2200 m [57,58]. In 10 studies, subjects were exposed to altitudes ranging from 3400 m to 4350 m [59,60,61,62,63,64,65,66,67,68]. The remaining six studies were performed at altitude levels ranging from 5050 m to 5700 m [69,70,71,72,73,74]. Altitude stays lasted from five to 22 days. In four studies, an ascent phase of four to eight days preceded the sojourn at altitude [64,71,72,73]. In the majority of the studies, subjects were adults, one study included children [64] whereas one study was conducted in the elderly [58]. Subjects remained physically active by daily hiking [58], trekking [64] or few ascents to higher altitudes [66] in four studies. After an initial fall in SpO_2_ in the beginning of the stay at altitude ≤ 2200 m (−3% to −6%), the subsequent increase in SpO_2_ during acclimatization ranged from 0% to 3% [57,75]. In studies performed at altitudes varying from 3400 m to 4350 m, the initial decline in SpO_2_ was reported to range from −5% to −19%. During acclimatization, SpO_2_ rose by 3% [62,63] in studies with short duration of exposure (5 days) and from 5%–8% in studies with longer duration (i.e., 6–22 days) [59,61,65,67,68]. When exercise was repeatedly performed during the stay at these altitudes, the increase in SpO_2_ over time ranged from 1%–3% [64,66]. One study that evaluated SpO_2_ data immediately before the start of an exercise test did not observe an acclimatization effect on SpO_2_ [60]. At altitudes > 5000 m, SpO_2_ initially decreased by −8% to −20%. SpO_2_ changes during acclimatization ranged from 6–8% [71,72,73]. In one study, SpO_2_ increased by 4% after 5 days at 5050 m but decreased again by 3% after 14 days [70]. Baillie et al., assessing AMS in 23 out of 35 subjects (66%), observed no acclimatization effect on SpO_2_ [74]. 

Changes in resting HR during the stay at altitude are mentioned in 10 out of the 20 studies. An initial increase in HR of 8 bpm at moderate altitude was observed in the study implementing daily hiking activities, with only small changes during acclimatization (−1 bpm) [58]. In the three studies performed at 3400 m to 3500 m, HR increased by 14–21 bpm, remained elevated when trekking exercise was performed during the stay [64] and decreased during acclimatization by −6 bpm after 5 days [62] and by −20 bpm after 9 days [60]. At altitudes >5000 m, the initial increases in HR ranged from 9 bpm to 25 bpm, whereas changes during acclimatization phase ranged from −1 to −9 bpm [69,70,71,72,73].

### 4.2. Exercising SpO_2_ and HR Changes during Acclimatization to High Altitude

Effects of acclimatization in SpO_2_ and HR during submaximal exercise are shown in Table 3. Studies were performed at altitudes ranging from 2800 m to 4300 m, except one study implementing exposures at 2000 m [75]. Altitude stays lasted from three to 22 days. In the studies with shorter duration of exposure (i.e., <7 days), initial fall in SpO_2_ after exercise ranged from −15% to −22%. In the end of the stay, SpO_2_ following exercise remained unchanged or increased by 3% [63,76,77]. In the studies lasting from seven to 22 days, SpO_2_ declined by −15% to −23% during exercise tests at acute hypoxia [60,66,68,78]. In these studies, the rise in SpO_2_ during acclimatization ranged from 2% to 10%.

Exercising HR at acute altitude exposure increased by 11 bpm to 38 bpm in six out of seven studies when compared to sea level. In the remaining two studies, reporting either 55% slower time trial durations [78] or significant reductions in mean power [77], exercising HR during acute altitude exposure declined in a range of −1 bpm to −7 bpm. In those studies, showing a primary increase in HR during the first exercise test, the reduction in exercising HR ranged from −4 bpm to −8 bpm [60,63,66,76], still being elevated compared to sea level. In the study performed at 2000 m, exercising HR returned to baseline after 7 days of active stay at altitude (−14 bpm) [58]. 

### 4.3. Changes in AMS Scores during Acclimatization to High Altitude 

The findings of 12 studies evaluating symptoms of acute mountain sickness during acclimatization using the Lake Louise Scoring System (LLS) are presented in Table 4. Reductions in SpO_2_ in these studies ranged from −5% to −29%. Altitudes ranged from 3180 m to 5400 m, and duration of exposures lasted between two to 11 days. In the studies with resting SpO_2_ levels below 80%, LLS ranged from 2.7 to 6.0 [72,74,79,80,81,82,83]. In all other studies, resting SpO_2_ remained above 80% with LLS ranging from 1.4 to 3.9, except in one study where all subjects developed AMS showing a LLS of 5.3 [84]. In most studies, LLS decreased over time when SpO_2_ was improved. However, in one study, LLS increased from 0.6 to 0.9 in a group of non-AMS subjects [85]. The only study that reported AMS-C scores observed similar time-courses as in the studies using LLS to assess AMS [86].

## 5. Part 3: Discussion

### 5.1. Physiologic and Pathophysiologic Mechanisms Explaining Pulse Oximetric Measures When Acutely Exposed to High Altitude and during Acclimatization

The presented findings from the literature review demonstrate rather large variations of pulse oximetry measures (SpO_2_ and HR) during acute exposure and acclimatization to high altitude. This is not surprising as conditions (levels of altitude/hypoxia, type of ascent, extent of pre-acclimatization, physical activity levels, characteristics of study populations, etc.) are also considerably different between studies. Main conclusions derived from this review are, that particularly SpO_2_ levels decrease with acute altitude/hypoxia exposure and partly recover during acclimatization, with an opposite trend of HR, i.e., initial increase and slight decrease during acclimatization. In addition, AMS development is consistently associated with lower SpO_2_ values compared to individuals free from AMS.

#### 5.1.1. Resting SpO_2_ and HR Changes during Acclimatization to High Altitude

When acutely ascending to high altitude, the human organism is exposed to the reduced availability of oxygen (hypoxia) because of the decreasing barometric pressure and related partial pressure of oxygen (pO_2_). As a consequence, several physiological responses are initiated to counteract hypoxia and the associated risk to get sick [88,89,90]. Such responses are targeted to improve oxygen delivery to tissues including hyperventilation (hypoxic ventilatory response, HVR), hemoconcentration due to diuresis, and elevated cardiac output due to sympathetic activation.

During the first few days at high altitude there is, compared to acute exposure, a progressive increase in resting ventilation (ventilatory acclimatization) which is accompanied by an increase in the arterial oxygen partial pressure (PaO_2_) and a decrease in the alveolar partial pressure of carbon dioxide (pACO_2_) [88,91]. Ventilatory acclimatization is characterized by an initial decrease and a subsequent increase in resting SpO_2_ values, reaching a maximum between 4 to 8 days, at least at an altitude up to 4300 m [65,68]. Provided altitude levels are comparable, increase in SpO_2_ seems to be higher when the time for acclimatization is prolonged (e.g., >5 days). In contrast to passive stays at altitude, frequent trekking activities or other exercises at altitude may delay the recovery of SpO_2_. The initial SpO_2_ decline is more pronounced and acclimatization may take some more days at higher compared to lower altitudes. An example for changes of SpO_2_ values during acclimatization at two different altitudes is depicted in Figure 3. Moreover, larger SpO_2_ variation during the first days become narrower with progressing acclimatization indicating slightly different individual time courses.

Resting HRs are typically elevated during the first few days after acute ascent to high altitude showing a subsequent decrease with acclimatization. However, the acclimatization effect on HR may be counteracted if the stay involves physical exercises [64]. An example for changes of HR values during acclimatization at 3600 m is shown in Figure 4. In the healthy general population, individual HR variation is much more pronounced than that of SpO_2_ values, e.g., depending on age, sex, and fitness, requiring careful consideration of the individual baseline values. Therefore, including HR values to assess the progress of acclimatization is a valuable complement to the SpO_2_ measurements.

It is important to mention that the presented pulse oximetry data reflect the time course of acclimatization predominantly after rapid ascent to a certain level of altitude. Various pre-acclimatization strategies including the use of hypoxia chambers, staging or mountaineering activities at moderate altitudes may accelerate the acclimatization process at a given high altitude [65,92,93]. On the other hand, higher than normal susceptibility to hypoxia exposure or pre-existing illnesses may delay this process [94,95].

#### 5.1.2. Exercising SpO_2_ and HR Changes during Acclimatization to High Altitude

Changes of SpO_2_ and HR values when acutely ascending to high altitude are more pronounced during submaximal exercise when compared to resting conditions [96]. From rest until the anaerobic threshold, there is an almost linear increase in heart rate and minute ventilation with increasing workload, the slope being steeper at high than low altitude [97]. The SpO_2_ decline during exercise depends, at least partly, on the individual ventilatory response and will become less steep with ventilatory acclimatization. This is based on the fact that the PaO_2_ at altitude is on the steep portion of the oxyhemoglobin dissociation curve, and an only slight fall in PaO_2_, e.g., because of hypoventilation, results in a marked SpO_2_ reduction [98]. The SpO_2_ during submaximal exercise is lowest during the first days at altitude and improves with acclimatization. Compared to resting conditions (4 to 8 days), the increase of exercising SpO_2_ values plateaus after 2 to 3 weeks during acclimatization to high altitude [68,99]. Thus, as observed in the studies presented in this review and similarly to resting SpO_2_, the magnitude of improvement of exercising SpO_2_ may be less pronounced in studies with shorter duration. An example for changes of (submaximal) exercising SpO_2_ during acclimatization to 4300 m is shown in Figure 5. Exercising heart rates decrease with increasing SpO_2_ during acclimatization exhibiting a similar time course [93]. In contrast to the aerobic capacity (VO_2_max), the initially reduced submaximal endurance performance at acute altitude improves during 2 to 3 weeks of acclimatization, accompanied by characteristic changes in cardiorespiratory responses [99,100,101,102]. For instance, the study by Horstman et al. reported a 31% and 59% improvement on day 9 and 15 without considerable changes until day 22 [99]. It has to be mentioned, that relevant acclimatization effects, i.e., hemoconcentration, hyperventilation and associated improved oxygenation and submaximal exercise performance, even occur during the first days at high altitude [77].

#### 5.1.3. The Use of Pulse Oximetry for the Diagnosis of Acute Mountain Sickness (AMS)

Although the usefulness of pulse oximetry for the prediction and diagnosis of AMS is still debated [103,104,105], there is increasing evidence confirming that subjects developing AMS are more hypoxic (lower SpO_2_ values) than those who stay free from AMS, provided conditions (e.g., level of pre-acclimatization, health status, etc.) are comparable and measurement methods are appropriate [6,8,12,106,107]. Various studies reported certain SpO_2_ cut-off values differentiating between AMS+ and AMS- at a certain level of altitude. For instance, Mandolesi et al. measured SpO_2_ values of 85.4% vs. 87.7% for AMS+ vs. AMS- subjects at 3275 m, and 84.5% vs. 86.4% at 3647 m [108]. However, different conditions between individuals can mislead prediction, e.g., subjects taking aspirin before high-altitude exposure tolerated lower SpO_2_ values (with regard to suffering from headache at high altitude, 3480 m) than those who were pretreated with placebo (83% vs. 88%) [109]. Thus, one might conclude that, when individual pre-conditions are carefully considered, the use of non-invasive pulse oximetry provides a simple and specific indicator of inadequate acclimatization to high altitudes associated with the risk for developing AMS [108]. In this context, the studies incorporated in the present review clearly show that when SpO_2_ levels are improved during the stay at altitude, AMS scores decline. However, a limiting factor regarding these studies is, that a more detailed information about AMS scores (i.e., separate scores of those who got AMS and those who did not) is lacking in most studies.

## 6. Conclusions

The presented findings indicate that despite the existing large variability, the use of finger pulse oximetry at high altitude is an invaluable tool in the evaluation of the individual course of acclimatization to high altitude but also for the monitoring of AMS progression and treatment efficacy.

However, there is a lack of data on the measurement accuracy of pulse oximeters at high altitudes. Therefore, in order to allow for optimal preconditions to obtain reliable data, complying with some specific technical and usage-oriented recommendations may enhance the accuracy and precision and reduce the bias of the measurement results. Especially when going to high altitudes, besides accuracy and precision, other technical characteristics of the device such as maximum operating altitude, minimum operating temperature, or the capability to detect low perfusion should be considered. There are also indications that medical pulse oximeters have an increased accuracy and lower deviations in SpO_2_ readings below 90% compared to “low-cost” handheld devices. In parallel to the technical aspects, the measurement process itself becomes more error-prone with increasing height. Some specific pitfalls in pulse oximetry as illustrated in this review affect the measurement results, in particular at high altitudes (e.g., SpO_2_ saturation below 70% or considerably decreased PaO_2_). Therefore, and as no uniform standard for measuring protocols at high altitude exists, an examiner with sufficient experiences in the data interpretation should perform the measurements. Importantly, when interpreting SpO_2_ data regarding to acclimatization, one should keep in mind that varying conditions such as pre-acclimatization phases, ascent rates, accomplished altitude levels, extent of physical activity or prevalence of AMS during the altitude stay may have a considerable impact on these data.

## Figures and Tables

**Figure 1 sensors-21-01263-f001:**
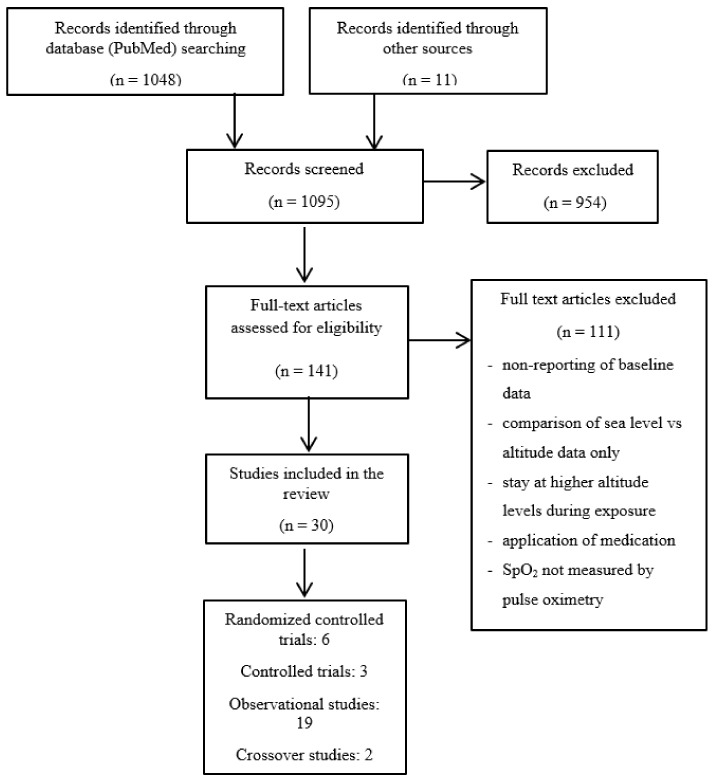
Flow chart of the study selection process.

**Figure 2 sensors-21-01263-f002:**
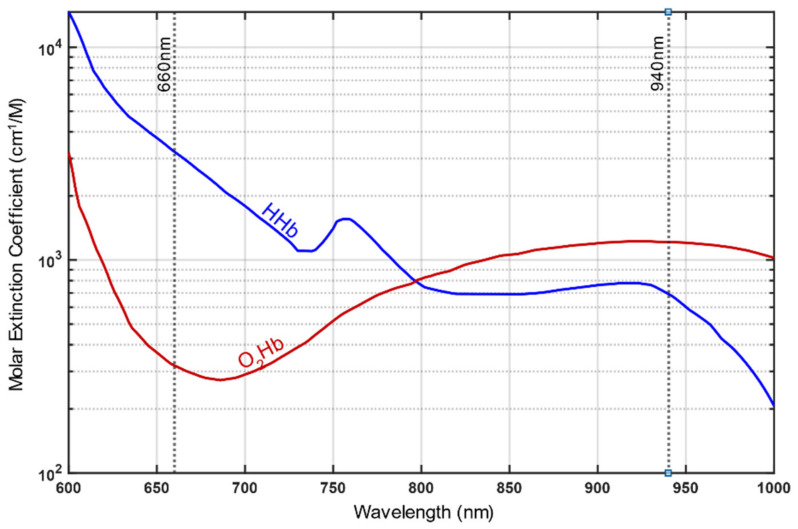
Light absorption spectrum of deoxyhemoglobin (HHb) and oxyhemoglobin (O_2_Hb). Different absorption for HHb and O_2_Hb at red light (660 nm) compared to infrared light (940 nm) is visible. (This Figure is based on data from Prahl, 1998 [20]).

**Figure 3 sensors-21-01263-f003:**
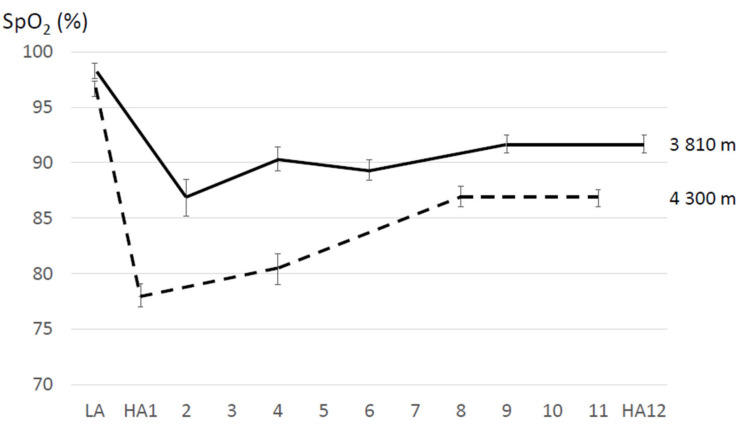
Example for changes of peripheral oxygen saturation (SpO_2_) when acutely ascending from low (LA) to high altitude (HA) and during the subsequent 11- or 12-day acclimatization period based on 2 studies performed at different altitudes (3810 m and 4300 m) [65,68]. At 3800 m, resting SpO_2_ was measured in a semi-supine position, with head and trunk elevated ~30°, by finger pulse oximetry (Criticare, 504-US pulse oxymeter). At 4300 m, resting SpO_2_ was measured in a sitting (upright) position for a 4-min period after relaxing for 20 min, by ear oximetry (Hewlett-Packard 47201A ear oximeter, Palo Alto, CA, USA).

**Figure 4 sensors-21-01263-f004:**
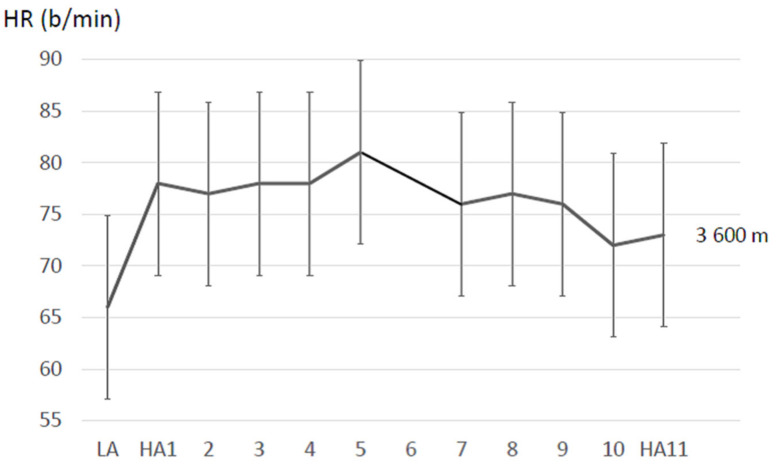
Example for changes of heart rate (HR) when acutely ascending from low (LA) to high altitude (HA1) and during the subsequent day acclimatization period at 3600 m based on a study with young soccer players (16 ± 0.4 years) [51]. HR data were collected in the morning after awakening with a Polar Team system (Polar Electro Oy, Kempele, Finland).

**Figure 5 sensors-21-01263-f005:**
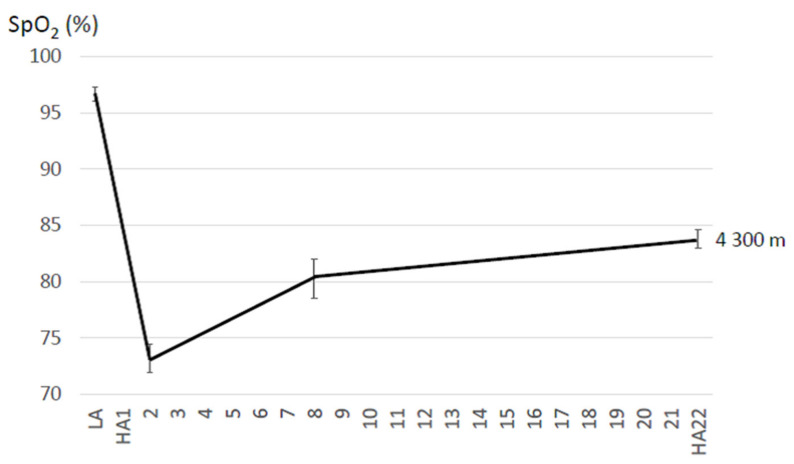
Example for changes of peripheral oxygen saturation (SpO_2_) during submaximal exercise when acutely ascending from low (LA) to high altitude (HA) and during the subsequent 22-day acclimatization period based on a study performed at 4300 m [68]. Resting SpO_2_ was measured in a sitting (upright) position for a 4-min period after relaxing for 20 min, by ear oximetry (Hewlett-Packard 47201A ear oximeter, Palo Alto, CA, USA).

**Table 1 sensors-21-01263-t001:** Most significant pitfalls and possible countermeasures for pulse oximetry particularly for healthy people at high altitudes.

Causes of Unreliable SpO_2_ Readings	Effects on The Measurement Result	Special Influence Conditioned by The High Altitude	Feasible Countermeasures
Excessive movement	Motion artefacts may cause a decrease of measured SpO_2_ [13,32,33], however modern devices implement advanced algorithm to reduce motion artefacts [14,15,17,33,34]. These devices may be identified by indications such as “motion tolerant” or “motion resistant” [34,35,36].	With increasing altitude, the temperature drops. This may result in cold extremities and an increased shivering and affects the sensor position and the sensor signal.	During measurement, keep the measuring position steady and avoid too much trembling.
Poor probe positioning	The red or infrared light of the sensor may bypass the tissue or too high levels of ambient light hit the light-detector of the sensor [13,15]. This results in a wrong SpO_2_ reading.	None	An imperfectly fitting of the sensor should be avoided, and the sensor should always be used in the appropriate position. If these countermeasures do not achieve the desired results, measurement at a different site (e.g., earlobe or forehead) may be considered.
Excess ambient light	Excessive ambient light can lead to erroneous SpO_2_ readings. However, modern devices are capable of handling strong ambient light more effectively [13,37].	Especially snow-covered areas with high solar radiation can lead to increased ambient light intensity at the sensor.	Protect the sensor from sunlight (e.g., by covering the measuring site).
Decreased arterial pressures of oxygen (PaO_2_)	A decrease in PaO_2_ (<60 mmHg) results in a significant change in oxygen saturation where small variations of the pressure have a strong effect on the saturation [12,38,39].	Increasing altitude results in a decreasing PaO_2_. Especially at altitudes above 3000 m a PaO_2_ below 60% can be expected [12,40,41].	To reduce fluctuations in PaO_2_, SpO_2_ measurements should be conducted after the person keeps silent and gentle breathing for several minutes. The measurement duration should be extended over a few minutes and the most frequent occurring value should be used [27].
SpO_2_ saturation below 70%	Devices complying with the international standard ISO 80601-2-61 (medical electrical equipment. Part 2-61: Specific requirements for basic safety and essential performance of pulse oximeter equipment) must measure accurate oxygen saturations (A_rms_ ≤ 4%) between 70–100%. Below 70%, they are less reliable [11,12,16,23,25,42].	the oxygen saturation is estimated based on human calibration data measured from 100% to 70%. Saturation values below 70% are only based on an extrapolation of this determined curve [13,25]. At high altitudes, however, the occurrence of low saturation values is not abnormal.	The possibility of a slight deviation of the measured value should be considered if oxygen saturation is below 70%. Especially when comparing data with devices of several manufacturers.
Cold-induced vasoconstriction (poor perfusion)	Cold skin temperatures reduce SpO_2_ reading accuracy [43,44,45]. This effect is based on a reduced blood flow due to cold-induced vasoconstriction [12,31]. However, modern devices can handle this condition and/or report it to the user. These devices may be identified by indications such as “oximeter with perfusion index” or “sensitivity to low perfusion signals”.	With increasing altitude, the temperature drops.	Warming the measuring site before and preferably during the measurement (e.g., using heating pads).
Skin pigmentation	Pulse oximeters are possibly less accurate during hypoxia in dark-skinned individuals at lower saturation (<80%) resulting in overestimations [46,47]. Feiner et al. mentioned [48]: “*further study is needed to confirm these observations in the relevant populations.*” However, an actual study is consistent with Bickler et al. and Feiner et al. [49].	At high altitude, the occurrence of saturation values below 80% is not abnormal [12,50].	Until the scientific data is more definite, the possibility of a slight overestimation (about +2% [51]) of the measured value should be kept in mind when interpreting the data for a person with oxygen saturation below approx. 90% combined with a dark pigmentation of the skin.
Nail polish	Some fingernail polish can lower the SpO_2_ readings [48]. Previous studies, however, have shown that the variance is not clinically relevant using actual devices [51,52,53,54].	None	Especially with older devices, the nail polish should be removed to avoid variations in the measurement accuracy. However, the deviation in the SpO_2_ readings is less than 2% [53,54].
Limited knowledge of technology (devices) and data interpretation	A lack of knowledge regarding device application and interpretation of the measurement data can lead to incorrect conclusions [14,23].	Conditions at high altitudes complicate the use of the device and the accurate interpretation of the data [12,27,50].	As Tannheimer et al. [50] concluded it “*requires an experienced examiner who can include altitude anamnesis, clinical examination and mountaineering aspects in the overall assessment*” to avoid possible pitfalls during SpO_2_ measurement and interpretation on high altitudes.
Dyshemoglobins (carboxyhemoglobin and methemoglobin)	Based on an absorption of the red and infrared light, methemoglobin (MetHb) and carboxyhemoglobin (HbCO) cause SpO_2_ overestimation and mask serious hypoxia [14,23,48]. However, as already mentioned, certain multiple-wavelengths devices are capable of detecting dyshemoglobins.	Unlikely at high altitude, however, it can be a danger using a cooking stove in small, enclosed areas like tents. In the worst case, this can lead to carbon monoxide poisoning [12,55].	When using devices that are not capable of analyzing dyshemoglobins, possible carbon monoxide poisoning should be considered if the person has remained in a small, enclosed space for an extended period while a combustion process (e.g., a stove) has taken place. Symptoms of carbon monoxide poisoning may include headache, nausea and drowsiness. However, these symptoms are similar to those associated to altitude sickness [56].

**Table 2 sensors-21-01263-t002:** Changes in resting SpO_2_ and HR during acclimatization to high altitude.

Authors	Participants M/F; Age (Means ± SD or Median (Range))	Altitude	Exposure Time at Target Altitude (days)	Information about SpO_2_ Measurements	Pre-Acclimatization/Prolonged Ascent Phase	Type of Exposure [Stay, Ascents]	Change in Resting SpO_2_ (%) (Means ± SD or Median (Range)) d (Days), Bl (Baseline)	Change in Resting HR (bpm) (Means ± SD or Median (Range))
Gangwar et al. (2019) [59]	20 M; 22–25 years ^#^	3520 m	7	n.a.	-	stay	Bl: 99.0 * d1: 89.7 * d2: 90.5 * d3: 91.2 * d4: 92.0 * d5: 93.0 * d6: 94.2 * d7: 95.2 *	
Voutselas et al. (2019) [69]	8 M; 48.0 ± 9.2 years	5700 m	7	environment temperature: 0.6–5 °C	-	stay	d1: 83.0 ± 3.7 d2: 82.9 ± 4.3 d3: 84.3 ± 3.3 d4: 86.9 ± 1.6 d5: 87.1 ± 5.5 d6: 84.6 ± 4.4 d7: 89.4 ± 1.8	d1: 87.9 ± 9.7 d2: 87.1 ± 8.5 d3: 86.6 ± 11.9 d4: 84.8 ± 10.5 d5: 95.1 ± 13.8 d6: 81.4 ± 9.3 d7: 78.8 ± 9.1
Gibson et al. (2015) [60]	29 (15 M, 14 F); 22.2 ± 5.4 years	3400 m	9	temperature (°C) (mean (95%CI)): Bl: 14.7 (14.1–15.4) d2: 24.2 (24.2–24.2) d6: 27.0 (27.0–27.0) d9: 21.9 (21.9–21.9)	-	stay/outdoor tests (6MWT) at 3 times at 42, 138 and 210 h	Bl: 97.7 (96.6–98.7) d2: 92.5 (91.6–93.3) d6: 91.6 (90.8–92.4) d9: 92.2 (91.4–93.0)	Bl: 82.0 (75.7–88.3) d2: 100.5 (95.8–105.2) d6: 85.1 (95.8–105.2) d9: 80.5 (75.4–85.7)
Hoiland et al. (2015) [70]	20 (15 M, 5 W); 34 ± 7 years	5050 m	14 (max 21)	n.a.	-	stay	Bl: 98.6 ± 1.1 d2: 79.5 ± 2.9 d5: 83.4 ± 1.9 d14–21: 80.5 ± 1.6	
Strapazzon et al. (2015) [61]	19 (15 M, 4 F); 39 ± 9 years	3830 m	8	SpO_2_ measured after signal stabilization; subjects at rest and with warm hands (SpO_2_: average of three consecutive measurements)	-	stay	Bl: 98.6 ± 1.4 9 h: 86.2 ± 5.8 d1: 87.1 ± 4.6 d2: 89.2 ± 3.9 d3: 91.5 ± 2.2 d8: 91.5 ± 3.1	Bl: 62.1 ± 8.0 9 h: 86.9 ± 18.4 d1: 82.2 ± 11.1 d2: 79.3 ± 14.7 d3: 76.3 ± 14.7 d8: 79.0 ± 10.8
Willie et al. (2014) [71]	8 (M,F); 28 ± 6 years	5050 m	14	SpO_2_ measured in triplicate, after 10 min rest in prewarmed sleeping bag (subjects: warm and calm before measurement)	1week at 1338 m; 6–8 day trek from 2860 m–5050 m (incl. 1day at 3440 m, 1–3day at 4371 m)	stay	Bl: 99.0 ± 0.3 d2: 80.0 ± 0.9 d8: 82.0 ± 0.9 d14: 86.0 ± 0.7	Bl: 58 ± 3 d2: 76 ± 4 d8: 75 ± 6 d14: 75 ± 5
Bhaumik et al. (2013) [62]	6 M; 24.8 ± 2.9 years	3500 m	5	subjects rested quietly in supine position; ambient temperature varied between 10–20 °C	-	stay	Bl: 98.3 ± 0.2 d2: 92.8 ± 0.5 d5: 96.5 ± 0.2	Bl: 67.0 ± 3.8 d2: 81.2 ± 4.1 d5: 75.7 ± 6.9
Agostoni et al. (2011) [73]	33 (22 M, 11 F); 40.8 ± 10.4 years	5400 m	14	experiments were performed in a heated tent	9 day ascent	stay	Bl: 97.6 ± 0.6 d1–2: 77.2 ± 6.0 d14–15: 85.3 ± 3.6	Bl: 73 ± 13 d1–2: 82 ± 19 d14–15: 77 ± 18
Fulco et al. (2011) [63]	9 (8 M, 1 F); 25 ± 6 years ^#^	4300 m	5	subjects rested in a seating position for 30 min; temperature was maintained at 21 ± 3 °C	-	stay	Bl: 97 ± 1 d1/d2: 82 ± 4 d5: 85 ± 5	
Modesti et al. (2011) [72]	47 (32 M, 15 F); 40 ± 9 years	5400 m	9–11	tests were carried out in a heated tent	2 day hike from 3440 m to 4200 m; 1 day stay at 4200 m; 2 day hike to 5400 m	stay	Bl: 98 ± 1 d1: 78 ± 6 d9–11: 86 ± 4	Bl: 61 ± 12 d1: 84 ± 16 d9–11: 78 ± 15
Baillie et al. (2009) [74]	42 (26 M, 16 F); 22.4 ± 6.3 years	5200 m	7	n.a.	4 day acclimatization at 3800 m	stay	Bl: 98 ± 1.3 d1: 77 ± 8 d3: 75 ± 5 d7: 77 ± 7	
Beidleman et al. (2009) [57]	11 M; 21 ± 3 years	2200 m	6	testing was performed in a climatically controlled room (temperature: 22 ± 2.8 °C)	-	stay	Bl: 97 ± 2 d1: 94 ± 1 d3: 93 ± 2 d6: 94 ± 2	Bl: 69 ± 6 d1: 68 ± 10 d2: 67 ± 10 d3: 66 ± 3
Scrase et al. (2009) [64]	9 (5 M, 4 F); 8 (6–13) years	3500 m	9	n.a.	4 day ascent 1300 m–3500 m	trekking (up to 3860 m)	Bl: 98.5 ± 0.9 d1: 88.9 ± 2.4 d9: 91.8 ± 1.5	Bl: 78 ± 13 d1: 99 ± 14 d9: 98 ± 14
Burtscher et al. (2001) [58]	20 (10 M, 10 F); 63.7 + 7.4 years	2000 m	7	10 min rest in a sitting position before measurement; HR and SpO_2_ measured continuously for 3 min and averaged over 15 s intervals (mean of the intervals in the final minute was taken as rest value)	-	daily hiking; 2.5 h (day1)- 5 h (day6); 50% VO_2_max	Bl: 96 ± 2 d1 (PM): 89.7 * d2 (AM/PM): 91.3/91.5 * d3 (AM/PM): 93.2/91.7 * d4 (AM/PM): 93.6/92.5 * d5 (AM/PM): 93.3/92.5 * d6 (AM/PM): 93.6/92.7 *	Bl: 60 ± 7 d2: 68.5 * d3: 70.5 * d4: 69.8 * d5: 69.9 * d6: 67.8 *
Sato et al. (1994) [65]	6 M	3810 m	12	n.a.	-	stay	Bl: 98.6 ± 0.37 d2: 86.2 ± 2.3 d4: 90.3 ± 1.1 d6: 89.4 ± 0.9 d9: 91.9 ± 0.6 d12: 91.0 ± 0.6 (means ± SEM)	
Savourey et al. (1994) [66]	7 (6 M, 1 W)	4350 m	7	n.a.	-	during stay: 3 ascents to Mont Blanc (4807 m)	Bl: 98 * d1: 85.0 (SEM 0.5) d7: 86.0 (SEM 0.7)	
Reeves et al. (1993) [67]	37 M	4300 m	19	n.a.	-	stay	Bl: 97 * d1: 81.0 ± 0.9 d2: 83 * d3: 85 * d4: 85 * d5: 86 * d7: 87 * d10: 88 * d19: 87.9 ± 0.4	
Bender et al. (1989) [68]	6 M; 21 ± 1 (mean ± SEM) years	4300 m	22	4-min measurement period; subjects sat upright after relaxing for at least 20 min	-	stay	97 * d1: 78.4 ± 1.6 d8: 87.5 ± 1.4 d20: 86.4 ± 0.6	

* data obtained from Figure. ^#^ control group. n.a. not available.

**Table 3 sensors-21-01263-t003:** Changes in exercising exercising SpO_2_ and HR during acclimatization to high altitude.

Authors	Participants M/F; Age (Mean ± SD or Median (Range)	Altitude	Exposure Time at Target Altitude (days)	Information about SpO_2_ Measurements	Type of Exercise Test	Change in Exercise SpO_2_ (%) (Means ± SD or Median (range)) d (Days), Bl (Baseline)	Change in Exercise HR (bpm) (Means ± SD or Median (Range))
Bradbury et al. (2020) [78]	6 M; 26.6 ± 8.5 years ^#^	4300	22	n.a.	80 min of metabolically-matched treadmill walking (2-mile time trial)	Bl: 95 ± 3 d1: 73 ± 4 d22: 81 ± 4 (data are means during 80 min time trial)	Bl: 175 ± 9 d1: 168 ± 14 d22: 161 ± 18 (data are means during 80 min time trial)
Gibson et al. (2015) [60]	29 (15 M, 14 F); 22.2 ± 5.4 years	3400 m	9	temperature [°C] (mean (95%CI)): Bl: 14.7 (14.1–15.4) d2: 24.2 (24.2–24.2) d6: 27.0 (27.0–27.0) d9: 21.9 (21.9–21.9)	6MWT	Bl: Pre: 97.7 (96.6–98.7) Post: 98.0 (97.5–98.6) d2: Pre: 92.5 (91.6–93.3) Post: 83.5 (81.8–85.2) d6: Pre: 91.6 (90.8–92.4) Post: 86.7 (85.2–88.2) d9: Pre: 92.2 (91.4–93.0) Post: 85.4 (83.7–87.2) (Pre/post exercise test)	Bl: Pre: 82.0 (75.7–88.3) Post: 116.3 (103.4–129.2) d2: Pre: 100.5 (95.8–105.2) Post: 154.7 (147.2–162.1) d6: Pre: 85.1 (95.8–105.2) Post: 148.1 (138.5–157.7) D9: Pre: 80.5 (75.4–85.7) Post: 149.1 (143.0–155.3) (Pre/post exercise test)
Burtscher et al. (2014) [76]	7 (4 M, 3 F); 44.7 ± 8.6 years ^#^	3480 m	3	SpO2 and HR were continuously monitored	3 min step test (stepping 90 times up and down; 4 cm step)	Bl: 95.2 ± 1.5 d1: 74.9 ± 5.9 d2: 76.1 ± 4.1 d3: 74.4 ± 3.7	Bl: 125 ± 12 HAd1: 144± 14 HAd2: 140 ± 12 HAd3: 140 ± 12
Fulco et al. (2011) [63]	9 (8 M, 1 F); 25 ± 6 years ^#^	4300 m	5	n.a.	20 min steady -state exercise at 45± 5% of SL VO_2_peak. (speed: 5.6 m/h)	Bl: 97 ± 1 d1: 75 ± 4 d2: 75 ± 4 d5: 78 ± 4	Bl: 129 ± 18 d1: 140 ± 15 d2: 138 ± 15 d5: 132 ± 12
Burtscher et al. (2006) [77]	5 M; 51.4 ± 7.7 years ^#^	2800 m	3	SpO_2_ was determined 5 times (minute 9, 19, 29, 39, 49); room temperature: ∼24 °C;	50 min cycle ergometer test at individually chosen power output	Bl: 94.2 ± 0.8 d1: 79.2 ± 3.2 d3: 82.1 ± 2.1	Bl: 167.6 ± 4.5 d1: 166.2 ± 5.1 d3: 164.2 ± 5.0
Burtscher et al. (2001) [60]	20 (10 M,10 F); 63.7 + 7.4 years	2000 m	7	SpO_2_ and HR measured continuously and averaged over 15-sec intervals; means of the intervals of the final minute indicate exercise responses	Step test (step up and down on a 24 cm-high step, 90 times in 3 min)	Bl: 93.2 ± 2.0 d1: 84.9 ± 2.8 d4 (AM): 88.1 ± 2.1	Bl: 124.3 ± 20.3 d1: 138.6 ± 19.2 d4 (AM): 124.7 ± 6.8
Savourey et al. (1994) [66]	7 (6 M, 1 F)	4350 m	7	n.a.	moderate cycle ergometer exercise at a constant power (100 W)	Bl: 98 * d1: 79.0 (SEM 1.8) d7: 82.0 (SEM 1.3)	Bl: 115 * d1: 135 * d7: 130 *
Bender et al. (1989) [68]	6 M; 21 ± 1 years	4300 m	22	n.a.	submaximal cycle exercise	d2: 72.7 d8: 78.6 d22: 82.3 *(means of measurements at min 5, 15 and 30)*	d2: (5 min): 155 ± 3 d2: (15 min): 157± 5 d2: (30 min): 150± 3 d8: (5 min): 162 ± 2 d8: (15 min): 165 ± 4 d8: (30 min): 159 ± 4 d22: (5 min): 168 ± 2, d22: (15 min): 169 ± 4 d22: (30 min): 163 ± 4

* data obtained from Figure. ^#^ control group. n.a. not available. No pre-acclimatization phases or prolonged ascent phases are included.

**Table 4 sensors-21-01263-t004:** Changes in AMS scores during stay at high altitude.

Authors	Participants M/F; Age (Men ± SD or Median (Range)	Altitude	Exposure Time at Target Altitude (days)	Information about SpO_2_ Measurements	Pre-Acclimatization/Prolonged Ascent Phase	Change in Resting SpO_2_ (%) (Means ± SD or Median (Range)) d (Days), h (Hours), Bl [Baseline]	Change in resting HR (bpm) (Means ± SD or Median (Range))	AMS (Lake Louise Score)
Vizcardo-Galindo et al. (2020) [79]	22 (21 M, 1 F); 32.7 ± 1.9 years	4340 m	4	n.a.	-	Bl: 98 * 12 h: 77 * 24 h: 77 * 36 h: 76 * 48 h: 81 * 72 h: 81 *	Bl: 72 * 12 h:80 * 24 h: 86 * 36 h: 89 * 42 h: 85 * 72 h: 84 *	Bl: 0 * 12 h: 1.2 * 24 h: 2.7 * 36 h: 1.3 * 42 h: 1.3 * 72 h: 0.8 *
Sareban et al. (2020) [87]	38 M; 19 endurance athletes: 31 ± 7 years 19 untrained: 38 ± 9 years	3450 m	2	SpO_2_ measured after rest in supine position for 10 min (stable SpO_2_ values were reached)	-	athletes: Bl: 97 ± 1 3 h: 82 ± 6 8 h: 81 ± 6 24 h: 87 ± 3 34 h: 85 ± 4 48 h: 87 ± 4 untrained: Bl: 96 ± 1 3 h: 83 ± 4 8 h: 83 ±4 24 h: 85 ± 5 34 h: 84 ± 4 48 h: 86 ± 3	athletes: Bl: 52 ± 9 3 h: 59 ± 8 8 h: 59 ± 11 24 h: 64 ± 8 34 h: 58 ± 9 48 h: 60 ± 8 untrained: Bl: 58 ±9 3 h: 72 ± 12 8 h: 68 ± 12 24 h: 76 ± 11 34 h: 69 ± 9 48 h: 73 ± 11	athletes: Bl: 0.1 * 3 h: 1.7 * 8 h: 2.1 * 24 h: 1.7 * 34 h: 0.7 * 48 h: 0.7 * untrained: Bl: 0.2 * 3 h: 1.2 * 8 h: 0.9 * 24 h: 1.7 * 34 h: 1.0 * 48 h: 1.0 *
Gekeler at al. (2019) [80]	14 (7 M, 7 F); 35 ± 8 years	4559 m	4	SpO_2_ measured after >5 min at rest after 1 min of steady recording	ascent: 1635 m to 4559 m within 24 h	Bl: 98.6 ± 1.3 d1 PM: 69.4 ± 4.4 d2 AM: 72.1 ± 5.9 d2 PM: 74.4 ± 7.1 d3 AM: 73.9 ± 6.0 d3 PM: 79.9 ± 5.4 d4 AM: 79.4 ± 4.3	Bl: 57.9 ± 7.0 d1 PM: 88.4 ± 6.0 d2 AM: 83.43 ± 10.1 d2 PM: 82.7 ± 9.5 d3 AM: 77.1 ± 12.2 d3 PM: 75.2 ± 16.6 d4 AM: 73.6 ± 13.4	Bl: 0 d1 PM: 5.4 ± 2.2 d2 AM: 5.4 ± 2.6 d2 PM: 3.9 ± 2.1 d3 AM: 4.0 ± 3.4 d3 PM: 2.1 ± 1.5 d4 AM: 2.4 ± 2.1
Lundeberg et al. (2018) [84]	9 (4 M, 5 F); 32.7 ± 11.7 years ^#^	3800 m	2.5	recordings lasted ~300 s; pulse oximeter was always placed on the same finger	-	Bl: 97.5 * h0: 85.5 * h12: 87.5 * h36: 89.5 * h60: 88.0 *	Bl: 70 * h0: 82 * h12: 82 * h36: 85 * h60: 85 *	h0: 2.6 * h12: 5.3 * h36: 3.4 * h60: 3.5 *
Gibson et al. (2015) [60]	29 (15 M, 14 F); 22.2 ± 5.4 years	3400 m	9	temperature [°C] (mean (95%CI)): Bl: 14.7 (14.1–15.4) d2: 24.2 (24.2–24.2) d6: 27.0 (27.0–27.0) d9: 21.9 (21.9–21.9)	-	Bl: 97.7 (96.6–98.7) d2: 92.5 (91.6–93.3) d6: 91.6 (90.8–92.4) d9: 92.2 (91.4–93.0)	Bl: 82.0 (75.7–88.3) d2: 100.5 (95.8–105.2) d6: 85.1 (95.8–105.2) d9: 80.5 (75.4–85.7)	Bl: 0.8 (0.4–1.1) d2: 2.0 (1.1–2.9) d6: 1.0 (0.3–1.6) d9: 1.0 (0.4–1.6)
Strapazzon et al. (2015) [61]	19 (15 M, 4 F); 39 ± 9 years	3830 m	8	SpO_2_ measured after signal stabilization, subject at rest and with warm hands (SpO_2_: average of three consecutive measurements)	-	Bl: 98.6 ± 1.4 9 h: 86.2 ± 5.8 d1: 87.1 ± 4.6 d2: 89.2 ± 3.9 d3: 91.5 ± 2.2 d8: 91.5 ± 3.1	Bl: 62.1 ± 8.0 9 h: 86.9 ± 18.4 d1: 82.2 ± 11.1 d2: 79.3 ± 14.7 d3: 76.3 ± 14.7 d8: 79.0 ± 10.8	Bl: 0.0 (0.0) 9 h: 0.6 (0.7) 24 h: 1.7 (2.7) 48 h: 0.5 (0.7) 72 h: 0.4 (0.5) d8: 0.1 (0.4)
Staab et al. (2013) [86]	18 M; 25 ± 5 years,	4300 m	3	data collection of at least 10 min; mean over the last 5–8 min of the session was calculated and used in the analyses; room temperature: 21 ± 2 °C	-	Bl: 99 ± 1 24 h: 81± 5 48 h: 83 ± 6 72 h: 83 ± 5	Bl: 52 ± 7 24 h: 75 ± 9 48 h: 75± 12 72 h: 74 ± 10	*AMS-C:* Bl: 0.2 * 24 h: 1.4 * 48 h: 1.4 * 72 h: 0.7*
Aeberli et al. (2012) [81]	25 (15 M, 10 F); 43.8 ± 9.5 years	4559 m	4	n.a.	1 night at 3650 m	Bl: 97.4 ± 1.5 d2: 78.4 ± 6.0 d4: 81.6 ± 8.6		Bl: 0.9 ± 1.0 d2: 3.4 ± 1.4 d4: 2.3 ± 1.3
Nussbaumer-Ochsner et al. (2012) [82]	16 (13 M, 3 F); 45 (33–50) years	4559 m	3 nights (n)	SpO_2_: mean value during sleep	-	Bl: 96 (95,96) n1: 67 (64,69) n3: 71 (69,78) *(medians and quartiles)*	Bl: 56 (50,61) n1: 81 (74,92) n3: 84 (75,89)	Bl: 1 (0,1) n1: 6 (3,9) n3: 4 (3,5)
Modesti et al. (2011) [72]	47 (32 M, 15 F) 40 ± 9 years	5400 m	9–11	tests were carried out in a heated tent	2 day hike from 3440 m to 4200 m; 1 day stay at 4200 m; 2 day hike to 5400 m	Bl: 98 ± 1 d1: 78 ± 6 d9–11: 86 ± 4	Bl: 61 ± 12 d1: 84 ± 16 d9–11: 78 ± 15	Bl: 0 d1: 2.7 ± 2.0 d9–11: 0.6 ± 1.0
Willmann et al. (2011) [83]	18 (11 M, 7 F); 35 ± 8 years	3647 m	4	SpO_2_ measurement performed after >5 min of rest	ascent within 24 h from 1635 m to 4559 m	Bl: 98.5 ± 1.3 d1 PM: 70.6 ± 5.2 d2 AM: 73.0 ± 6.0 d2 PM: 74.3 ± 6.6 d3 AM: 74.2v5.6 d3 PM: 77.3 ± 3.8 d4 AM: 79.6 ± 5.2	Bl: 60.3 ± 7.2 d1 PM: 89.4 ± 5.9 d2 AM: 82.7 ± 9.7 d2 PM: 80.1 ± 11.7 d3 AM: 76.8 ± 11.4 d3 PM: 75.3 ± 13.2 d4 AM:72.2 ± 12.7	Bl: 0 d1 PM: 5.7 ± 3.1 d2 AM: 5.2 ± 2.6 d2 PM: 3.8 ± 2.0 d3 AM: 4.1 ± 3.0 d3 PM: 2.1 ± 1.4 d4 AM: 2.3 ± 1.9
Baillie et al. (2009) [74]	42 (26 M, 16 F); 22.4 ± 6.3 years	5200 m	7	n.a.	4 day acclimatization at 3800 m	Bl: 98 ± 1 d1: 77 ± 8 d3: 75 ± 5 d7: 77 ± 7		d1: 4 (2–5) d2: 4 (2–6) d3: 3 (1–5) d7: 1 (0–2) *median (IQR)*
Chen et al. (2008) [85]	27 (11 M, 16 F); 39 ± 12 years	3180 m	2 nights (n)	measurements performed with the subjects in a supine position after resting for 10 min	-	*AMS+* (N = 13) Bl: 97.7 ± 0.8 4–6 h: 85.5 ± 3.2 n2: 87.3 ± 2.1 *Non-AMS:* Bl: 97.4 ± 1.1 4–6 h: 85.5 ± 4.3 n2: 87.1 ± 4.4	*AMS +:* Bl: 63.5 ± 6.5 4–6 h: 77.7 ± 7.8 n2: 79.1 ± 11.3 *Non-AMS:* Bl: 64.5 ± 8.2 4–6 h: 75.0 ± 9.1 n2: 80.1 ± 9.3	*AMS +:* Bl: 0.0 ± 0.0 4–6 h: 3.9 ± 2.4 n2: 2.9± 2.7 *Non-AMS:* Bl: 0.0 ± 0.0 4–6 h: 0.6 ± 0.9 n2: 0.0 ± 0.0

* data obtained from Figure. ^#^ control group. N.a. not available.

## Data Availability

All available data are provided.

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
