# Peer review of "The Use of Pulse Oximetry in the Assessment of Acclimatization to High Altitude"

_sensors, 2021, doi:10.3390/s21041263_

Round 1

Reviewer 1 Report

The aim of this ms is “providing information on the functioning of pulse oximeters, appropriate measurements methods and published time courses of pulse oximetry data….during [acute] exposure to various altitudes.”  Lines 46-49 (see also Abstract lines 17-19)

These points are intended to encourage revisions to improve the value of the ms.  This reviewer’s practical experience with pulse oximeters prompts some of the points.

MAJOR POINTS

The authors review two bodies of literature.  The first deals with pulse oximeters themselves and the second deals with their use during acute exposure to high-altitude hypoxia.  The review would be vastly improved if the information about pulse oximeters were used to evaluate the quality and credibility of the information about oxygen saturation changes during acute exposure to high altitude.  That would integrate the two bodies of literature and might tell us something new. 

Part 2.1. provides the basic principles of pulse oximetry along with relevant pitfalls and possible countermeasures.  This is useful as far as it goes and would become more useful if the authors explained why different pulse oximeters give different results when measured at the same individual at the same time?   For example, could it be the number of wavelengths?  Or the use of different calibration curves?  When pulse oximeters register O2Hb, are the hemoglobin molecules fully saturated with four O2 molecules or partially with 1 – 3 (but not zero)?   Does this review include both finger pulse oximetry and that obtained at other sites such as the forehead?

TABLE 2 effectively illustrates the variety of exposure protocols, the small sample sizes.  To integrate the two themes of this review, please add a column to this table that describes the extent to which the mentioned studies considered the possible confounding factors identified in Part 2.1.  Basic Principles.   With that information, the authors can identify more and less reliable studies among those they reviewed.  Then they can use the more reliable to address the questions about acclimatization.

Part 2.3.1.  Discussion. The conclusion (lines 215 – 220) tells us something we have known for many decades; it will strike most readers as nothing new and nothing that warrants a publication.  The conclusion does not mention the technical aspects of oximetry or the benefits of appropriate protocols. 

The last sentence of the ms states “However, the use of appropriate pulse oximetry devices, measurement protocols and sufficient experiences win the data interpretation are prerequisites for proper assessment.”   The authors do not offer evidence to support this statement.  They could, perhaps, if they evaluated the reviewed studies based on the criteria mentioned in that last sentence. 

OTHER CONSIDERATIONS AND SUGGESTIONS:

Abstract:  line 15 states that ‘some controversy remains’.  What is that controversy and where is it mentioned in the presentation, how does that influence interpretation of the study results?

Figure 1.  Does this come from a person, a sample of people, hemoglobin in test tube?

Some of the recommendations would likely insult readers unless rephrased or accompanied with further information.  An example is a gratuitous remark on lines 122-123 “…technologically up-to-date equipment is recommended to be used.”   Please tell the reader how to apply this recommendation to evaluate oximeters.

Please specify what you mean by ‘high accuracy’ and ‘less accurate’ and ‘slight overestimation’.  E.g. line 108 and Table 1, row on skin pigmentation.

Review English language usage to remove odd word usage.  For instance, look at Table  1.  The heading to column 3 says ‘large height’ instead of high altitude.

TABLE 1

Row 2, how does one identify a new generation device? 

Row 3,” poor probe position” can result in the failure of a measurement (so may many of these pitfalls).  Consider adding small fingers (e.g. children) or arthritic fingers that result in a light leak around the fingers or the probes’ failure to close adequately.  Cold extremities may be warmed.  A countermeasure is to use the correct sized finger sensor or measure a different site (if this review includes oximetry at all sites).

Row 5 . “Increased partial pressure of oxygen”.   This row doesn’t make sense as presented now.  Do the authors really mean decreased?  Perhaps this row really refers to day-to-day variation in barometric pressure or, alternatively, what happens when someone hyperventilates (that occurs at altitudes and during pregnancy)?   

Row 6, “SpO2 saturation below 70%”.  What is ISO 80601-2-61 in column 2?   Columns 3 and 4 touch on this reviewer’s question about various devices but doesn’t tell us what to do after we ‘question data reliability” and “find deviation”. 

Row 7 “Hypothermia vasoconstriction” (often called cold-induced vasoconstriction).  Column 2 should mention that some oximeters report finger perfusion measurements. 

Row 8 “Skin pigmentation”.  Column 2: How should we identify and quantify ‘dark-skinned individuals”?  Should skin reflectance be measured and reported?   

Row 9 “modern technology”.  How do we know what we are using?

Row 11 ‘Dyshemoglobins” Some pulse oximeters report metHb and COhb.  Smokers will have elevated CoHb.  Generators emit CO.    Last column:  how should we decide if a measured saturation is ‘unremarkable’?

TABLE 2, column 7 “Change in resting SpO2”.   Are the values means or medians?  Are the +/- SDs or SEMs?  What are the values in parentheses?

TABLE 3.  The acclimatization protocols are extremely varied.  Here again, please add a column evaluating the extent to which the authors took possible confounders into consideration.   And what about previous exposure to high-altitude hypoxia?

TABLE 4.    Readers really want to know the scores of those who got AMS and those who did not.    

FIGURES 2 and 3.   The word exemplary in the legends is misleading.  Perhaps the authors intended to use the word example?    To what extent do the studies summarized in these figures take into account the technical aspects of pulse oximetry.    How young are the young soccer players plotted in Figure 3? 

Author Response

Reviewer 1:

Dear Reviewer,

first of all, thank you very much for the really helpful and constructive review. We did our best in addressing all your comments (see below).  Changes made in the revised manuscript are highlighted.

MAJOR POINTS

Comment 1:

The authors review two bodies of literature.  The first deals with pulse oximeters themselves and the second deals with their use during acute exposure to high-altitude hypoxia.  The review would be vastly improved if the information about pulse oximeters were used to evaluate the quality and credibility of the information about oxygen saturation changes during acute exposure to high altitude.  That would integrate the two bodies of literature and might tell us something new.

Response to comment 1:

We fully agree and thank the reviewer for these suggestions. However, unfortunately, the information given about the application of the pulse oximeters (and how studies considered possible confounding factors) is very scarce or even missing and thus does not allow a reliable evaluation of the quality and credibility of the SpO2 changes. Nevertheless, we included a new column in all tables showing the available information about SpO2 measurements. Hopefully, this has led at least to some improvement expected by the reviewer.

Comment 2:

Part 2.1. provides the basic principles of pulse oximetry along with relevant pitfalls and possible countermeasures.  This is useful as far as it goes and would become more useful if the authors explained why different pulse oximeters give different results when measured at the same individual at the same time?  

  • For example, could it be the number of wavelengths?
  • Or the use of different calibration curves?
  • When pulse oximeters register O2Hb, are the hemoglobin molecules fully saturated with four O2 molecules or partially with 1 – 3 (but not zero)?
  • Does this review include both finger pulse oximetry and that obtained at other sites such as the forehead?

Response to comment 2:

Thank you for these useful comments. Normal devices that comply with the ISO standard should have a maximum deviation of 4% from the SaO2 value. This information and the main technical reasons for possible deviations in the measurements have now been added in the revised version. Devices that do not comply with this standard (non-medical, low-cost) now have also been considered. Based on published studies, those are now compared with the medical devices.

  • “For example, could it be the number of wavelengths?”:

Multiple wavelengths are used in CO-oximetry to detect dyshemoglobins in addition to oxy- and deoxyhemoglobin. If dyshemoglobins are present at a normal level, the results of a "normal" pulse oximeter should be highly correlated with multiple wavelength devices. However, this is certainly an interesting point for the reader, we have included this information in the revision. Thank you for this comment.

  • “Or the use of different calibration curves?”:

This could of course be a reason for small deviations. Unfortunately, we do not know if each manufacturer uses marginally different calibration curves. Therefore, this was not mentioned in the article.

  • “When pulse oximeters register O2Hb, are the hemoglobin molecules fully saturated with four O2 molecules or partially with 1 – 3 (but not zero)?”:

Saturation depends on partial pressure of oxygen (PO2). The higher the pressure, the sooner the hemoglobin is completely saturated. If the pressure drops (under normal conditions below 60 mmHg), the hemoglobin saturation drops sharply (oxygen-hemoglobin dissociation curve). Below 60mmHg, even small changes have a large effect on saturation. This fact is indicated in the table (row " Decreased arterial pressures of oxygen").

  • “Does this review include both finger pulse oximetry and that obtained at other sites such as the forehead?”:

Mainly, the sensors are positioned on the finger. A study by Ross et al. even shows that at high altitudes devices that measure at the finger show the best results. However, other common locations such as the earlobe or forehead can be used for measurements during movement. This information (with additionally references) has now been provided for the reader in the recommendations for selecting a device.

Comment 3:

TABLE 2 effectively illustrates the variety of exposure protocols, the small sample sizes.  To integrate the two themes of this review, please add a column to this table that describes the extent to which the mentioned studies considered the possible confounding factors identified in Part 2.1.  Basic Principles.   With that information, the authors can identify more and less reliable studies among those they reviewed.  Then they can use the more reliable to address the questions about acclimatization.

Response to comment 3:

Thank you again. Please, see our response to comment 1.

Comment 4:

Part 2.3.1.  Discussion. The conclusion (lines 215 – 220) tells us something we have known for many decades; it will strike most readers as nothing new and nothing that warrants a publication.  The conclusion does not mention the technical aspects of oximetry or the benefits of appropriate protocols.

Response to comment 4:

We thank the reviewer very much for this hint. We changed the whole paragraph focusing on the technical aspects of oximetry and handling of the device. This paragraph reads as follows:

The presented findings indicate, despite the existing large variability, that the use of finger pulse oximetry at high altitude is an invaluable tool in the evaluation of the individual course of acclimatization to high altitude but also for the monitoring of AMS progression and treatment efficacy. However, there is a lack of data on the measurement accuracy of pulse oximeters at high altitudes. Therefore, in order to allow for optimal preconditions to obtain reliable data, complying with some specific technical and usage-oriented recommendations may enhance the accuracy and precision and reduce the bias of the measurement results.  Especially when going to high altitudes, besides accuracy and precision, other technical characteristics of the device such as maximum operating altitude, minimum operating temperature, or the capability to detect low perfusion should be considered. There are also indications that “medical” pulse oximeters have an increased accuracy and lower deviations in SpO2 readings below 90% compared to “low-cost” handheld devices. In parallel to the technical aspects, the measurement process itself becomes more error-prone with increasing height. Some specific pitfalls in pulse oximetry as illustrated in this review affect the measurement results, in particular at high altitudes (e.g., SpO2 saturation below 70% or considerably decreased PaO2). Therefore, and as no uniform standard for measuring protocols at high altitude exists, an examiner with sufficient experiences in the data interpretation should perform the measurements. Importantly, when interpreting SpO2 data with regard to the acclimatization status, one should keep in mind that varying conditions such as pre-acclimatization phases, ascent rates, accomplished altitude levels, extent of physical activity or prevalence of AMS during the altitude stay may have a considerable impact on these data.

Comment 5:

The last sentence of the ms states “However, the use of appropriate pulse oximetry devices, measurement protocols and sufficient experiences with the data interpretation are prerequisites for proper assessment.”   The authors do not offer evidence to support this statement.  They could, perhaps, if they evaluated the reviewed studies based on the criteria mentioned in that last sentence.

Response to comment 5:

We removed this sentence from the conclusion as the available data does not allow such an evaluation (please see response to comment 1). We therefore changed this paragraph (please see response to comment 4).

OTHER CONSIDERATIONS AND SUGGESTIONS:

Comment 6:

Abstract:  line 15 states that ‘some controversy remains’.  What is that controversy and where is it mentioned in the presentation, how does that influence interpretation of the study results?

Response to comment 6:

In the abstract we refer to the “controversies” that are now well readable from the tables. In the introduction section of the manuscript, we also refer to a study that provides further details about the existing controversy.

Comment 7:

Figure 1.  Does this come from a person, a sample of people, hemoglobin in test tube?

Response to comment 7:

These data were extracted by Scott Prahl (https://omlc.org/spectra/hemoglobin/summary.html) and includes multiply sources. These data are also used in numerous other publications.  In our description, these data are only used for illustration purposes to explain the reader why these 2 specific wavelengths have been chosen.

Comment to comment 8:

Some of the recommendations would likely insult readers unless rephrased or accompanied with further information.  An example is a gratuitous remark on lines 122-123 “…technologically up-to-date equipment is recommended to be used.”   Please tell the reader how to apply this recommendation to evaluate oximeters.

Response to comment 8:

Thank you for this reasonable comment. However, it is difficult to provide the reader recommendations to evaluate oximeters suitable for high altitudes.  Devices usually only specify a range of SpO2 saturation from 70% to 100%, additionally the maximum operating altitude and operating temperature is sometimes not designed for extreme altitudes and the associated low temperatures. Additionally, how a pulse oximeter determines the values below 70% is also not published by the manufacturer (these values can only be extrapolated mathematically). A feasible way would be to test pulse oximeters under study conditions at high altitude and compare the results with SaO2 measurements (blood gas analysis). Unfortunately, we could only find a limited number of studies that explicitly test pulse oximeters for their functionality at high altitudes and possibly even compare the results to SaO2. However, in order to provide the reader with a reference point, we have included recommendations for the selection of a device. Additionally, a new table listing portable and handheld devices that have been used in relevant studies since 2011 was added.

Comment 9:

Please specify what you mean by ‘high accuracy’ and ‘less accurate’ and ‘slight overestimation’.  E.g., line 108 and Table 1, row on skin pigmentation.

Response to comment 9:

Thank you! This has now been specified in more detail.

Comment 10:

Review English language usage to remove odd word usage.  For instance, look at Table  1.  The heading to column 3 says ‘large height’ instead of high altitude.

Response to comment 10:

During revision of the manuscript and tables, another proper language proof reading was performed.

Comment 11:

TABLE 1

Responses to comments (11) on table 1:

Row 2, how does one identify a new generation device?

Response: Thank you for that comment. In the literature, such devices are described as having improved motion correction and better detection in less perfused tissues. By now, many manufacturers should have implemented this. The user can recognize this as the manufacturer labels these devices such as "motion tolerant” or “motion resistant". To avoid misunderstandings, we now refer to “modern devices” instead of “new generation devices” and the corresponding points have been specified.

Row 3,” poor probe position” can result in the failure of a measurement (so may many of these pitfalls).  Consider adding small fingers (e.g., children) or arthritic fingers that result in a light leak around the fingers or the probes’ failure to close adequately.  Cold extremities may be warmed.  A countermeasure is to use the correct sized finger sensor or measure a different site (if this review includes oximetry at all sites).

Response: This useful hint was implemented.

Row 5 . “Increased partial pressure of oxygen”.   This row doesn’t make sense as presented now.  Do the authors really mean decreased?  Perhaps this row really refers to day-to-day variation in barometric pressure or, alternatively, what happens when someone hyperventilates (that occurs at altitudes and during pregnancy)?  

Response: This information (in that row) was concretized.

Row 6, “SpO2 saturation below 70%”.  What is ISO 80601-2-61 in column 2?  

Response: This refers to the international Standard for medical electrical equipment. Part 2-61: Specific requirements for basic safety and essential performance of pulse oximeter equipment. This information was added.

Columns 3 and 4 touch on this reviewer’s question about various devices but doesn’t tell us what to do after we ‘question data reliability” and “find deviation”.

Response: The information has been specified in column 2 and therefore removed from column 3. Important for the reader is that due to the extrapolation below 70%, different manufacturers can deliver slightly different results. Therefore, care must be taken when comparing values of different devices. This info is contained in column 4.

Row 7 “Hypothermia vasoconstriction” (often called cold-induced vasoconstriction).  Column 2 should mention that some oximeters report finger perfusion measurements.

Response: Thank you. A notice about this technology has been added.

Row 8 “Skin pigmentation”.  Column 2: How should we identify and quantify ‘dark-skinned individuals”?  Should skin reflectance be measured and reported?  

Response: This “quantification” is not clear from the data. However, a new actual study concerning this topic was added and the data in the table were adjusted.

Row 9 “modern technology”.  How do we know what we are using?

Response: Since "older studies" (2003) could not find any clinical significance. The term was changed to " actual devices". However, we cannot conclude to all devices (especially for non-clinical devices). However, the reader's attention should be drawn to the fact that slight discrepancies may occur. Therefore, the countermeasure nail polish should be avoided, as there is a possibility of a slight deviation. This has been concretized.

Row 11 ‘Dyshemoglobins” Some pulse oximeters report metHb and COhb.  Smokers will have elevated CoHb.  Generators emit CO.   

Response: The information of “advanced” multiple wavelength oximeters was added. However, this article does not focus on smokers (car-exhaust inhalation and prolonged exposure to heavy-traffic environments can also probably be ruled out) at high altitudes. The danger CO of during a combustion process has been pointed out.

Last column:  how should we decide if a measured saturation is ‘unremarkable’?

Response: This has been changed and the reader's attention is now drawn to the fact that CO2 poisoning should be considered if symptoms are present.

Comment 12:

TABLE 2, column 7 “Change in resting SpO2”.   Are the values means or medians?  Are the +/- SDs or SEMs?  What are the values in parentheses?

Response to comment 12:

We thank the reviewer for pointing that out! Now, we added this information in all tables.

Comment 13:

TABLE 3.  The acclimatization protocols are extremely varied.  Here again, please add a column evaluating the extent to which the authors took possible confounders into consideration.   And what about previous exposure to high-altitude hypoxia?

Response to comment 13:

We also added a column showing the available information about the measurement of SpO2 in table 3. In the legend of table 3, we included the information that no pre-acclimatization phases or prolonged ascent phases are included in the studies presented in this table.

Comment 14:

TABLE 4.    Readers really want to know the scores of those who got AMS and those who did not.   

Response to comment 14:

In table 4 we present all AMS data that are available in the particular studies. Unfortunately, information about the scores of those who got AMS and those who did not is lacking in most studies. This information has been added to the discussion section (5.1.3) now.

Comment 15:

FIGURES 2 and 3.   The word exemplary in the legends is misleading.  Perhaps the authors intended to use the word example? To what extent do the studies summarized in these figures take into account the technical aspects of pulse oximetry. How young are the young soccer players plotted in Figure 3?

Response to comment 15:

We changed the wording in the text of the manuscript and in the legend of figures 2 and 3 as suggested.

The following information on the available measurement methods, used devices, and the age of soccer players have been included:

Fig.2 (now fig.3): Example for changes of peripheral oxygen saturation (SpO2) when acutely ascending from low (LA) to high altitude (HA) and during the subsequent 11- or 12-day acclimatization period based on 2 studies performed at different altitudes (3,810 m and 4,300 m) [29, 32]. At 3,800 m, resting SpO2 was measured in a semi-supine position, with head and trunk elevated ~ 30°, by finger pulse oximetry (Criticare, 504, US). At 4,300 m, resting SpO2 was measured in a sitting (upright) position for a 4-min period after relaxing for 20 min, by ear oximetry (Hewlett-Packard, 47201A).

Fig. 3 (now fig.4): Example for changes of heart rate (HR) when acutely ascending from low (LA) to high altitude (HA1) and during the subsequent day acclimatization period at 3,600 m based on a study with young soccer players (16 ± 0.4 yrs) [95]. HR data were collected in the morning after awakening with a Polar Team system (Polar Electro Oy, Kempele, Finland).

Reviewer 2 Report

Dear editors,

I appreciate the opportunity to review this manuscript. the revised topic is very interesting. there is controversy regarding the device and protocols; therefore the importance of this review. My comments are added below.

After careful reading of the manuscript, you could say that the methodology used does not correspond to the highly recommended PRISMA standards for reviews.

For example, the inclusion criteria considered are not very specific; Are they RCTs, observational studies? A flow chart is not provided. this information is important in reviews.

Data extraction could provide a statistical analysis (meta-analysis), which would increase the quality of the manuscript. Here's how the purpose of the review might be better answered:

Therefore, this review is aimed at providing information on the functioning of pulse oximeters, appropriate measurement methods and published time courses of pulse oximetry data (peripheral oxygen saturation, SpO2; and heart rate, (HR) recorded at rest and submaximal exercise during exposure to various altitudes.”

Lines 93, 110, 117. To review “see Error! Reference source not found”

Author Response

Reviewer 2:

Dear Reviewer,

first of all, thank you very much for the really helpful and constructive review. We did our best in addressing all your comments (see below).  Changes made in the revised manuscript are highlighted.

Comment 1:

After careful reading of the manuscript, you could say that the methodology used does not correspond to the highly recommended PRISMA standards for reviews.

For example, the inclusion criteria considered are not very specific; Are they RCTs, observational studies? A flow chart is not provided. this information is important in reviews.

Response to comment 1:

Thank you for these points. In the revised version of the manuscript, we included a flow chart as suggested by the reviewer. In addition, in the method section we give the information that all type of studies were included.

We understand the reviewer, but based on the primary aims and the available data we were unable to perform a systematic review and/or meta-analysis, and therefore, some of the PRISMA items, especially the risk of bias, summary measures and synthesis of results, could not be applied. Nevertheless, we did our best to include as many studies as possible (please, see the flow chart) and to provide a balanced view. Thanks for understanding.

Comment 2:

Data extraction could provide a statistical analysis (meta-analysis), which would increase the quality of the manuscript. Here's how the purpose of the review might be better answered:

“Therefore, this review is aimed at providing information on the functioning of pulse oximeters, appropriate measurement methods and published time courses of pulse oximetry data (peripheral oxygen saturation, SpO2; and heart rate, (HR) recorded at rest and submaximal exercise during exposure to various altitudes.”

Response to comment 2:

Yes, this was our initial intention, but unfortunately, we had to reject. Please, see also the response above. The heterogeneity of study aims and available studies (e.g., different study designs, varying altitude levels, performance levels of participants, duration of altitude stays, application of SpO2 measurements, type of implemented exercise), did not allow to perform a reliable meta-analysis. We would be happy to find your agreement.

Comment 3:

Lines 93, 110, 117. To review “see Error! Reference source not found”

Response to comment 3:

In the revised version of the manuscript provided by the editor, this error indication does not appear.

Thank you very much again!

Round 2

Reviewer 1 Report

Thank you for responding to my requests for clarification.   This paper will be a useful reference.

Reviewer 2 Report

Thanks to the authors for the response to the comments made.
The current version of the manuscript is considered suitable for publication.

Best regards.